# Diversity-oriented functionalization of 2-pyridones and uracils

Yong Shang[1,3], Chenggui Wu[1,2,3], Qianwen Gao[1], Chang Liu[1], Lisha Li[1], Xinping Zhang[1], Hong-Gang Cheng[1], Shanshan Liu [1] & Qianghui Zhou [1✉]

Heterocycles 2-pyridone and uracil are privileged pharmacophores. Diversity-oriented synthesis of their derivatives is in urgent need in medicinal chemistry. Herein, we report a palladium/norbornene cooperative catalysis enabled dual-functionalization of iodinated 2-pyridones and uracils. The success of this research depends on the use of two unique norbornene derivatives as the mediator. Readily available alkyl halides/tosylates and aryl bromides are utilized as *ortho*-alkylating and -arylating reagents, respectively. Widely accessible *ipso*-terminating reagents, including H/DCO$_2$Na, boronic acid/ester, terminal alkene and alkyne are compatible with this protocol. Thus, a large number of valuable 2-pyridone derivatives, including deuterium/CD$_3$-labeled 2-pyridones, bicyclic 2-pyridones, 2-pyridone-fenofibrate conjugate, axially chiral 2-pyridone (97% *ee*), as well as uracil and thymine derivatives, can be quickly prepared in a predictable manner (79 examples reported), which will be very useful in new drug discovery.

[1] Sauvage Center for Molecular Sciences, Engineering Research Center of Organosilicon Compounds & Materials (Ministry of Education), College of Chemistry and Molecular Sciences, and The Institute for Advanced Studies, Wuhan, China. [2] Key Laboratory of Xin'an Medicine, Ministry of Education, Anhui University of Chinese Medicine, Hefei, Anhui, China. [3] These authors contributed equally: Yong Shang, Chenggui Wu. ✉email: qhzhou@whu.edu.cn

2-Pyridone is an important class of electron-deficient heterocycle, widely found in bioactive natural products (Fig. 1A)[1–4], eg. (−)-maximiscin, (+)-hosieine A, and (+)-lyconadin A. Recently, 2-pyridones have even been utilized as pivotal ligands in transition metal catalysis[5,6]. More importantly, 2-pyridone unit is recognized as a privileged pharmacophore prevalent in pharmaceutical agents (Fig. 1A)[7–12], for instance, huperzine A (anti-Alzheimer)[9], milrinone (anti-heart failure)[10], SD-560 (anti-fibrosis)[11], and camptothecin (antitumor)[12]. The latest example is tazemetostat (Tazverik™), the first EZH2 inhibitor approved by FDA in early 2020 for epithelioid sarcoma treatment[13]. In addition, several promising 2-pyridone-based EZH2 inhibitors are in clinical trials[14,15]. As revealed by crystallographic studies[16,17], the 2-pyridone moiety acts as the common warhead of EZH2 inhibitors, thus it is crucial for their EZH2 enzyme inhibition activities (Fig. 1B). Following the success of tazemetostat, there is an urgent need from pharmaceutical industry for diversity-oriented synthesis (DOS)[18,19] of 2-pyridone derivatives library, therefore to find new generation of EZH2 inhibitors via high-throughput biochemical screening (HTS)[14,15].

To date, significant progresses have been made for the preparation of 2-pyridone derivatives (Fig. 2A)[20–23]. Besides the classical strategies regarding intrinsic electrophilic substitutions[24,25], pyridines hydrolysis[26], and de novo construction of the 2-pyridone ring from acyclic precursors[21,27,28], transition metal catalysis played an increasingly important role in developing step-economic methods[20–23]. Specifically, the catalytic site-selective C–H functionalization of 2-pyridones has become an emerging strategy[23,29–32]. However, these methods are

commonly single-tasked, thus the obtained 2-pyridones are of limited diversity[20–23]. Additionally, some of them usually require specially functionalized substrates[20–22,24–28,32] or harsh reaction conditions[21,27], significantly limiting their scopes. Therefore, development of general and efficient approaches for rapid synthesis of 2-pyridone derivatives library from readily available starting materials is a highly desirable yet challenging subject[14,15].

Palladium/norbornene (Pd/NBE) cooperative catalysis (namely, the Catellani reaction)[33] is recognized as a powerful strategy for expeditious synthesis of highly substituted arenes[34–41]. Owing to diversified and orthogonal dual-functionalization (ortho/ipso) of aryl halides (mainly aryl iodides), the Catellani reaction has become a versatile tool for quickly building aromatics library[39–41]. However, current scope of this chemistry is mainly limited to aromatic substrates, and its application to partially aromatic scaffolds is rarely reported[42–45], because the vinylic C–H bonds are generally more challenging to functionalize than aryl C–H bond[46–48]. In 2018, the Yamamoto group reported a unique two-component vinylogous Catellani annulation for the assembly of tricyclic benzo-fused products, involving partially aromatic 4-iodo-2-quinolones and 4-iodo-coumarin as substrates[43]. Later on, the Dong group successively realized two elegant alkenyl Catellani reactions, utilizing alkenyl (pseudo)halides[44] and alkenes with a directing group[45] as substrates, respectively. Inspired by their research, we envisaged to apply this challenging partially aromatic Catellani strategy for diversity-oriented synthesis of 2-pyridones. As shown in Fig. 2B, 4-iodo-2-pyridone (1), electrophilic alkyl/aryl halide (2), and terminating reagent (3) would engage in sequential ortho-C–H activation (intermediate I to II), functionalization (II–IV), and

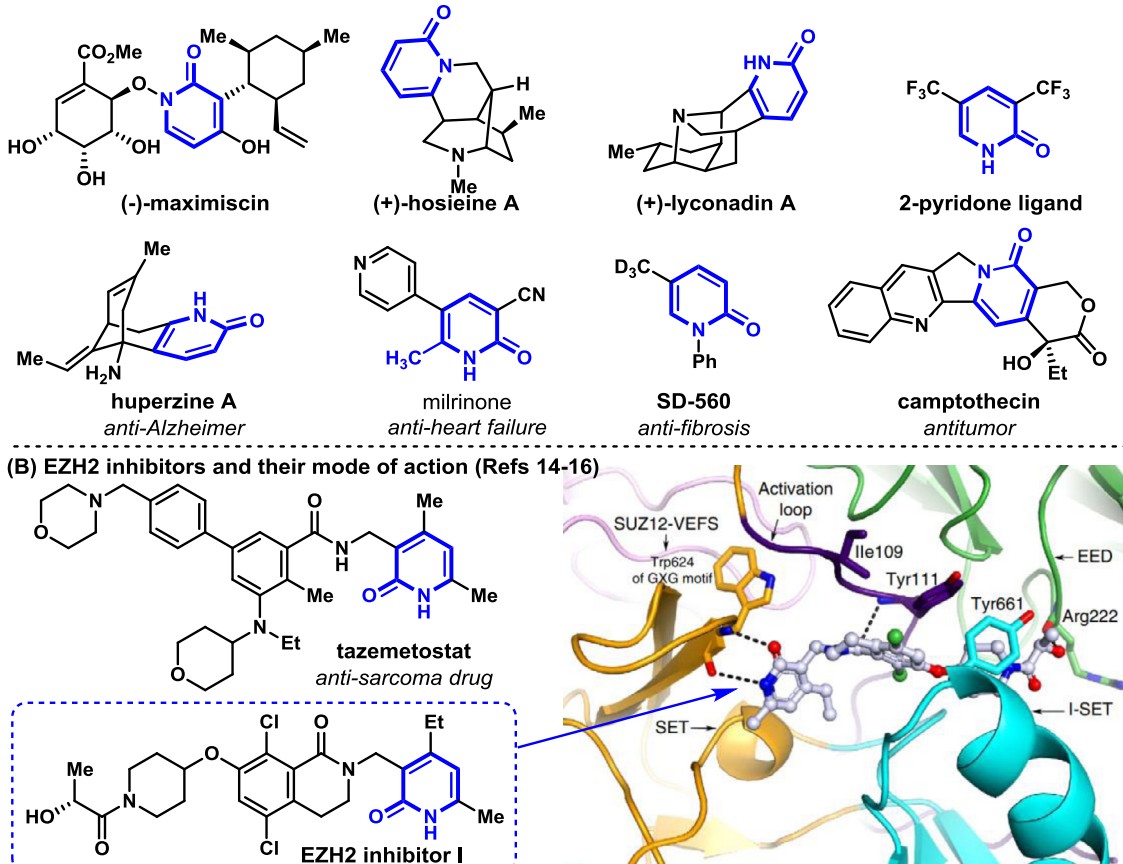

**(A) Structurally related natural products, ligands and pharmaceuticals**

(-)-maximiscin

(+)-hosieine A

(+)-lyconadin A

2-pyridone ligand

huperzine A
*anti-Alzheimer*

milrinone
*anti-heart failure*

SD-560
*anti-fibrosis*

camptothecin
*antitumor*

**(B) EZH2 inhibitors and their mode of action (Refs 14-16)**

tazemetostat
*anti-sarcoma drug*

EZH2 inhibitor I

**Fig. 1 2-Pyridone units in natural products, ligands, and pharmaceuticals. A** Structurally related natural products, ligands, and pharmaceuticals. **B** EZH2 inhibitors and their mode of action.

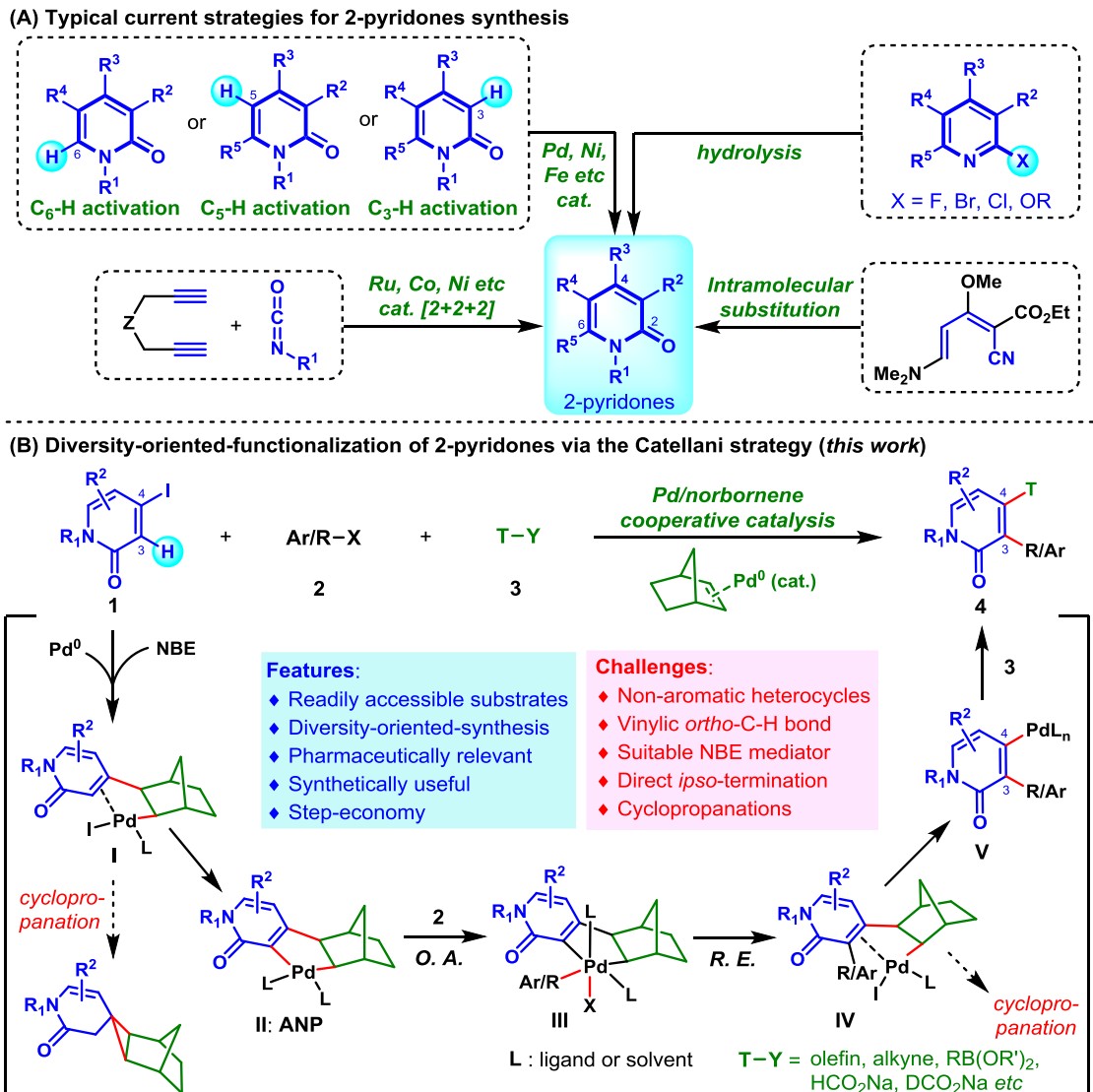

**Fig. 2 Strategies for 2-pyridones synthesis. A** Typical current strategies for 2-pyridones synthesis. **B** Diversity-oriented-synthesis of 2-pyridones via the Catellani strategy.

*ipso*-termination (intermediate **IV** to product **4**). Such a sequence would enable controllable formation of two vicinal chemical bonds in a single operation (good step-economy). Hence, this Catellani-type dual-functionalization strategy would generate structurally very diversified 2-pyridones, thereby meeting the increasing needs from medicinal chemists[14,15]. Although this hypothesis is mechanistically feasible, the following challenges are foreseeable. First, although the iodinated 2-pyridones are easily accessible[49], owing to their potential chelating ability[5,6], they are challenging substrates and have never been used in Catellani reactions before[34–41]. Second, several potential side reactions, eg. cyclopropanation (via intermediates **I** and **IV**, Fig. 2B)[43,44,47,48], direct coupling of **1** and **3** etc, will be competitive pathways. Third, a versatile NBE mediator needs to be identified, to promote the vinylic *ortho*-C–H activation and prevent the side reactions as well.

In this wok, we develop a palladium/norbornene cooperative catalysis-enabled diversity-oriented functionalization of heterocycles 2-pyridone and uracil. Readily available alkyl halides/ tosylate and aryl bromides are utilized as *ortho*-alkylating and -arylating reagents, respectively. Widely accessible H/DCO₂Na, boronic acid/ester, alkene and alkyne are employed as *ipso*-

terminating reagents. A large number of useful derivatives of these heterocycles, including deuterium/CD₃-labeled 2-pyridones, bicyclic 2-pyridones, 2-pyridone-fenofibrate conjugate, axially chiral 2-pyridone (97% *ee*), as well as uracil and thymine derivatives, can be quickly prepared in a predictable manner.

## Results and discussion

### Ortho-alkylation of 2-pyridones

*Reaction design and optimization.* Our efforts commenced with a model reaction, using 1-benzyl-4-iodo-5-methyl-2-pyridone (**1a**), ethyl 4-bromobutanoate (**2a**), and styrene (**3a**) as the reactants, to optimize the reaction conditions (see Supplementary Tables 1–5). Partial screening results were summarized in Table 1. The optimal catalyst, base, solvent, and temperature combination was Pd (OAc)₂ (5 mol%), K₂CO₃ (2.5 equiv), dioxane, and 105 °C. Interestingly, phophine ligand was demonstrated dispensable for this reaction (see Supplementary Table 3)[33,50–54]. Nevertheless, a significant mediator effect was observed. With the promotion of simple **N¹** (1.0 equiv), the reaction furnished the desired product **4a** in 27% yield (entry 1). Readily available norbornene derivatives[55–59] **N²−N⁶** showed similar reactivity as **N¹**, and

**Table 1 Optimization of reaction conditions[a].**

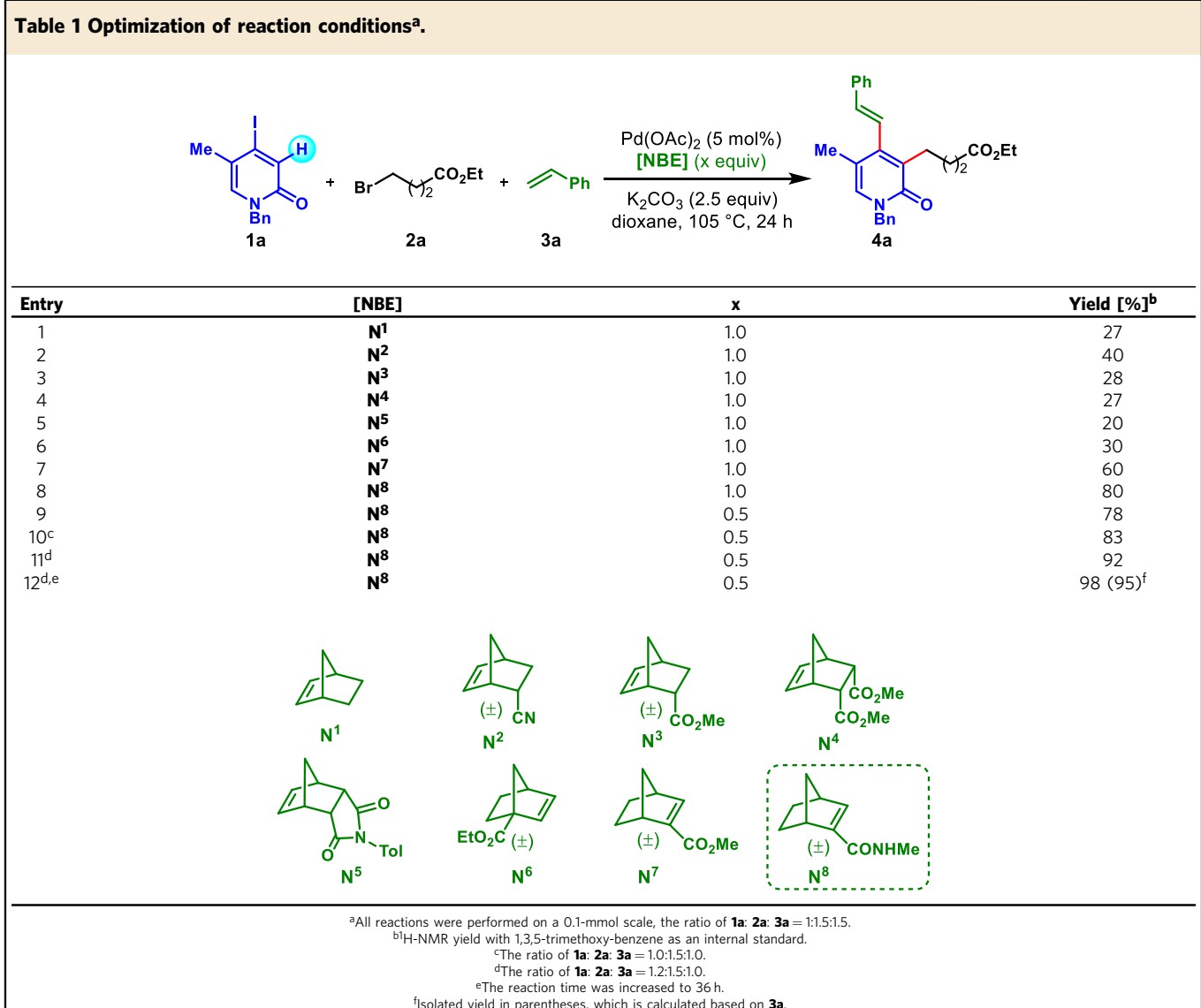

| Entry | [NBE] | x | Yield [%][b] |
|---|---|---|---|
| 1 | N[1] | 1.0 | 27 |
| 2 | N[2] | 1.0 | 40 |
| 3 | N[3] | 1.0 | 28 |
| 4 | N[4] | 1.0 | 27 |
| 5 | N[5] | 1.0 | 20 |
| 6 | N[6] | 1.0 | 30 |
| 7 | N[7] | 1.0 | 60 |
| 8 | N[8] | 1.0 | 80 |
| 9 | N[8] | 0.5 | 78 |
| 10[c] | N[8] | 0.5 | 83 |
| 11[d] | N[8] | 0.5 | 92 |
| 12[d,e] | N[8] | 0.5 | 98 (95)[f] |

[a]All reactions were performed on a 0.1-mmol scale, the ratio of **1a**: **2a**: **3a** = 1:1.5:1.5.
[b]1H-NMR yield with 1,3,5-trimethoxy-benzene as an internal standard.
[c]The ratio of **1a**: **2a**: **3a** = 1.0:1.5:1.0.
[d]The ratio of **1a**: **2a**: **3a** = 1.2:1.5:1.0.
[e]The reaction time was increased to 36 h.
[f]Isolated yield in parentheses, which is calculated based on **3a**.

delivered **4a** in 20−40% yields (entries 2–6). Gratifyingly, the Yu's mediator (**N[7]**)[56] significantly improved the reacton efficiency to deliver **4a** in 60% yield (entry 7), while the Dong's mediator (**N[8]**)[44] further increased the yield to 80% (entry 8). Thus, **N[8]** was identified as the optimal mediator (for additional mediator screening, see Supplementary Table 4). Further optimization indicated that the amount of **N[8]** could be reduced to 50 mol% without deleterious effects (entry 9). Since **1a** had multiple side reaction pathways, e.g., direct Heck reaction, deiodination, annulation with [NBE] etc, and **3a** (1.5 equiv) was always in excess after the reaction, we therefore changed the molar ratio of reactants (entries 10−11). Good results were obtained with the ratio of **1a**: **2a**: **3a** = 1.2:1.5:1.0, and yield of **4a** was increased to 92% (entry 11). Finally, the optimal conditions involved increasing the reaction time to 36 h, which furnished **4a** in 98% yield (95% isolated yield; entry 12).

*Substrate scope.* With the optimal reaction conditions confirmed, we first examined the scope of 2-pyridone, with bromide **2a** and styrene **3a** as the reaction partners. As shown in Table 2A, a series of 4-iodo-2-pyridones with substitution at C3 or C5 position, including methyl, fluoro, chloro, methyl ether, and ester group,

reacted smoothly to provide the alkylated products (**4a**−**4j**) in 60–95% yields. The *N*-substitution of 2-pyridones could be benzyl (**4a**), methyl (**4b**), 2,4,6-trimethylbenzyl (Mesityl, **4c**), methoxymethyl (MOM, **4d**), and p-methoxybenzyl (PMB, **4g**). Surprisingly, even the one with free N–H delivered the corresponding product **4e** in 69% yield with simultaneous nucleophilic *N*-alkylation. For *ortho*-unsubstituted 4-iodo-2-pyridone, the dialkylated product **4l** was obtained in 39% yield. Notably, after minor modification of the standard conditions, biologically relevant 6-iodo uracil also became a suitable substrate to afford the desired product (**4m**) in moderate yield. The practicality and robustness of this protocol are evident from the 3.0 mmol scale experiment, which led to a gram-scale preparation of product **4a** (1.22 g, 98% yield), alongside the recovery of mediator **N[8]** in 87% yield.

Then, the scope of alkylating reagent (**2**) was examined (Table 2B). Common methylation reagents including trimethylphosphate (**2b**)[60], iodomethane (**2c**)[61,62], methyl tosylate (**2d**)[60], and its deuterated sibling (**2e**)[60] were good reactants to deliver the (deuterated)methylation products (**4n**−**o**) in good to excellent yields (77−92%). Other simple alkylating reagents, eg. ethyl bromide (**2g**), butyl bromide (**2h**), and benzyl chloride (**2i**) gave the desired products (**4p**−**r**) in excellent yields (91−93%).

**Table 2 Reaction scope of 2-pyridone and alkylating reagent.**

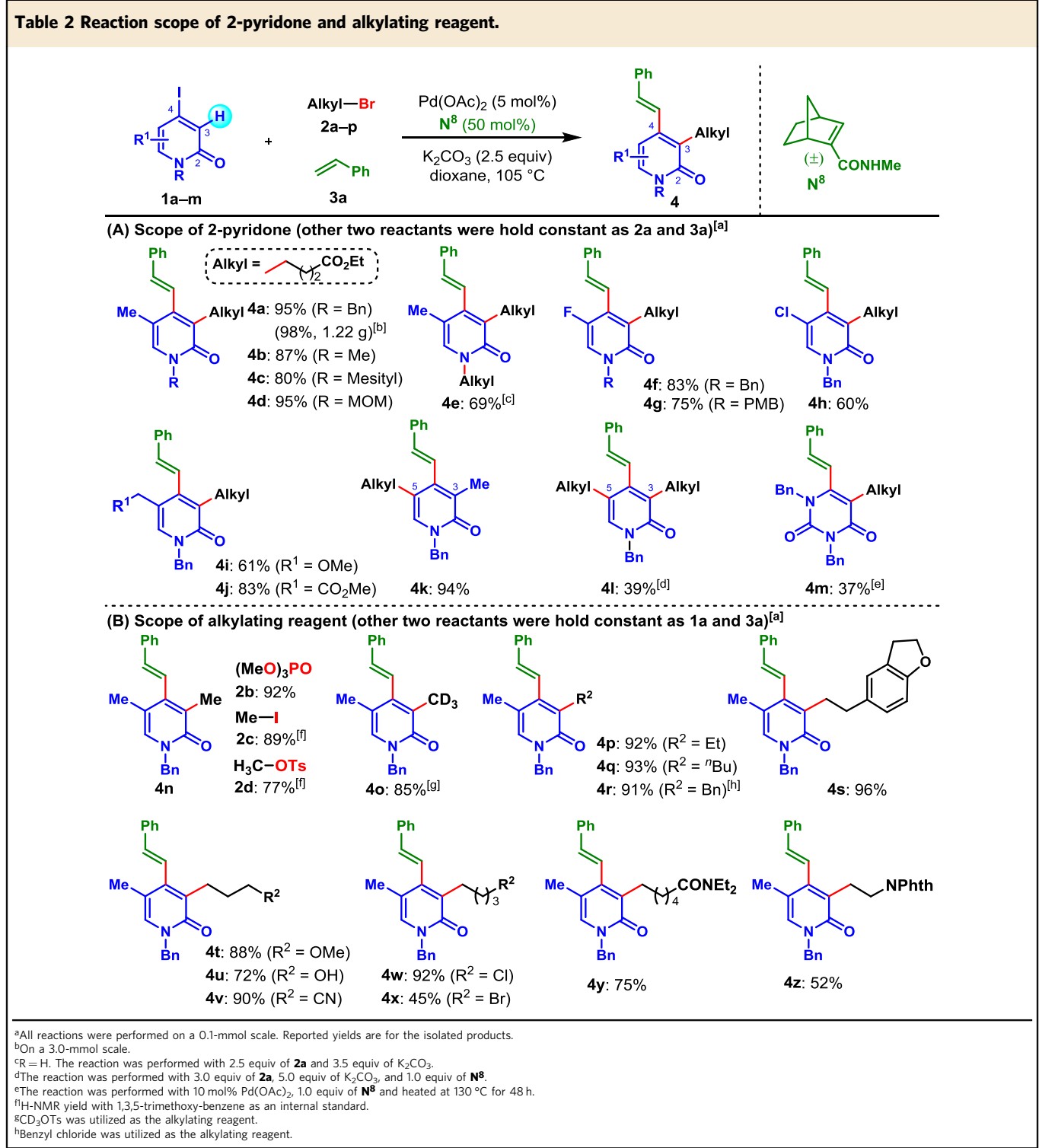

(A) Scope of 2-pyridone (other two reactants were hold constant as 2a and 3a)[a]

**4a**: 95% (R = Bn)
(98%, 1.22 g)[b]
**4b**: 87% (R = Me)
**4c**: 80% (R = Mesityl)
**4d**: 95% (R = MOM)

**4e**: 69%[c]

**4f**: 83% (R = Bn)
**4g**: 75% (R = PMB)

**4h**: 60%

**4i**: 61% (R¹ = OMe)
**4j**: 83% (R¹ = CO₂Me)

**4k**: 94%

**4l**: 39%[d]

**4m**: 37%[e]

(B) Scope of alkylating reagent (other two reactants were hold constant as 1a and 3a)[a]

(MeO)₃PO
**2b**: 92%
Me—I
**2c**: 89%[f]
H₃C—OTs
**2d**: 77%[f]

**4n**

**4o**: 85%[g]

**4p**: 92% (R² = Et)
**4q**: 93% (R² = ⁿBu)
**4r**: 91% (R² = Bn)[h]

**4s**: 96%

**4t**: 88% (R² = OMe)
**4u**: 72% (R² = OH)
**4v**: 90% (R² = CN)

**4w**: 92% (R² = Cl)
**4x**: 45% (R² = Br)

**4y**: 75%

**4z**: 52%

aAll reactions were performed on a 0.1-mmol scale. Reported yields are for the isolated products.
bOn a 3.0-mmol scale.
cR = H. The reaction was performed with 2.5 equiv of **2a** and 3.5 equiv of K₂CO₃.
dThe reaction was performed with 3.0 equiv of **2a**, 5.0 equiv of K₂CO₃, and 1.0 equiv of **N⁸**.
eThe reaction was performed with 10 mol% Pd(OAc)₂, 1.0 equiv of **N⁸** and heated at 130 °C for 48 h.
fH-NMR yield with 1,3,5-trimethoxy-benzene as an internal standard.
gCD₃OTs was utilized as the alkylating reagent.
hBenzyl chloride was utilized as the alkylating reagent.

Bromides containing an array of functional groups including methoxyl (**4t**), cyano (**4v**), chloro (**4w**), bromo (**4x**), amide (**4y**), and protected amino (**4z**) gave moderate to excellent yields (45−96%). Notably, free hydroxyl group (**4u**) was also compatible with this protocol.

Next, the scope of terminating reagent **3** was also explored. As shown in Table 3A, a wide range of olefins with electron-poor, -rich, or -neutral property were suitable substrates, including acrylates (**4a′** and **4e′**), acrylamide (**4b′**), vinyl phosphonic ester

(**4c′**), phenyl vinyl sulfone (**4d′**), vinylsilane (**4f′**), olefinic alcohol (**4g′**), and styrenes (**4h′−j′**), providing the desired 2-pyridones in excellent yields (80−92%). Notabley, the reaction of trimethyl (vinyl)silane proceeded to afford the desilylation product (**4f′**) in 82% yield. The reactions with a complex styrene derived from antihyperlipidemic drug medicine fenofibrate proceeded uneventfully to afford **4j′** in 90% yield. Besides olefins, other types of terminating reagents were also applicable under the standard reaction conditions (Table 3B), including H/DCO₂Na

**Table 3 Reaction scope with respect to the terminating reagent.**

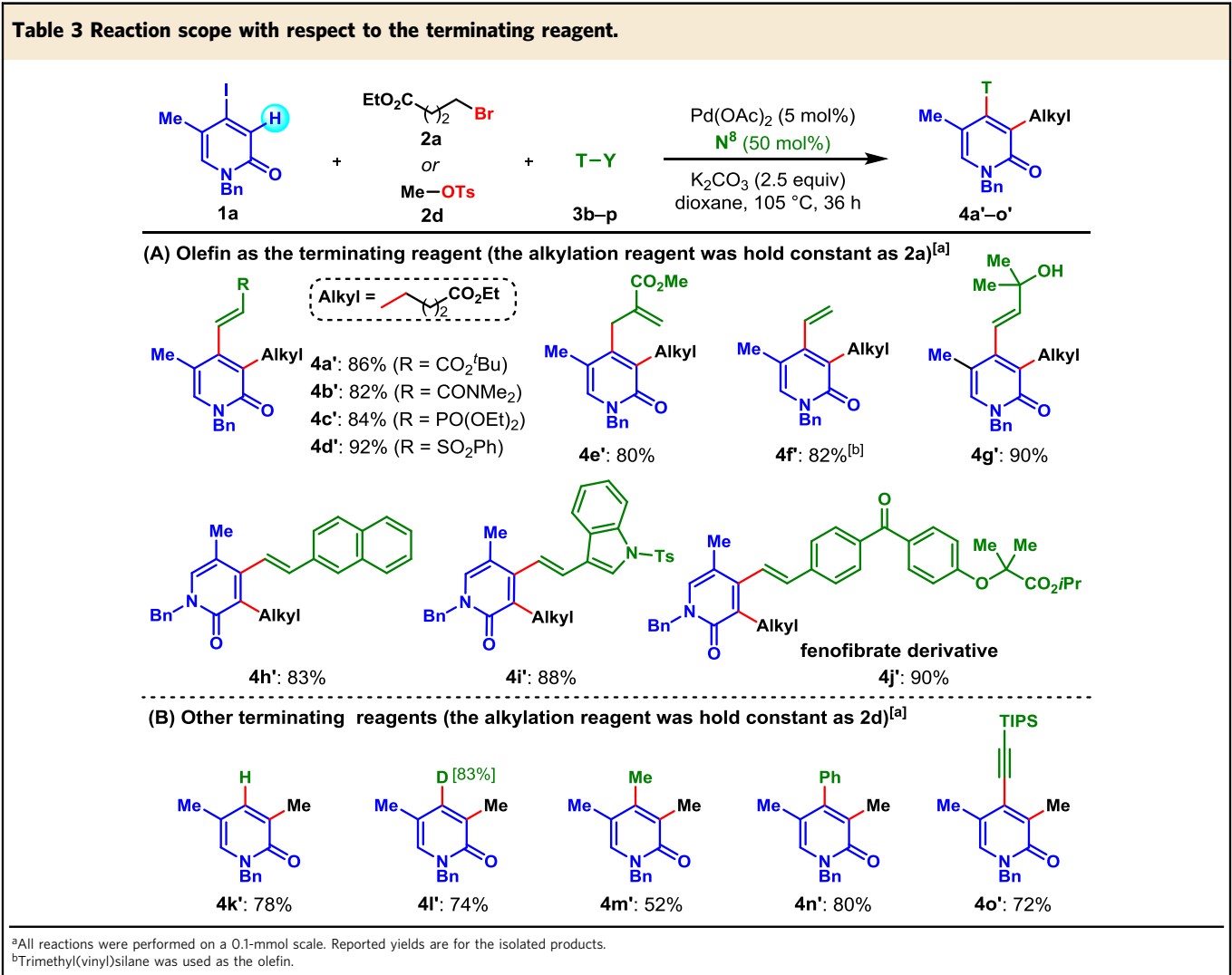

(A) Olefin as the terminating reagent (the alkylation reagent was hold constant as 2a)[a]

**4a':** 86% (R = CO₂ᵗBu)
**4b':** 82% (R = CONMe₂)
**4c':** 84% (R = PO(OEt)₂)
**4d':** 92% (R = SO₂Ph)

**4e':** 80%

**4f':** 82%[b]

**4g':** 90%

**4h':** 83%

**4i':** 88%

fenofibrate derivative
**4j':** 90%

(B) Other terminating reagents (the alkylation reagent was hold constant as 2d)[a]

**4k':** 78%

**4l':** 74%

**4m':** 52%

**4n':** 80%

**4o':** 72%

ᵃAll reactions were performed on a 0.1-mmol scale. Reported yields are for the isolated products.
ᵇTrimethyl(vinyl)silane was used as the olefin.

(**4k′** and **4l′**)[62,63], MeB(OH)₂ (**4m′**)[61], PhBpin (**4n′**), and terminal alkyne (**4o′**)[64], and the corresponding 2-pyridone products were obtained in 52−80% yields.

**Ortho-arylation of 2-pyridones.** Encouraged by the success of *ortho*-alkylation of 2-pyridones, we proceeded to explore the *ortho*-arylation. Gratifyingly, after minor modification of previous standard conditions, including the use of mediator **N⁹**[59] and a solvent change to DME (see Supplementary Tables 6-7), we were able to achieve this goal. Notably, the loading of Pd(OAc)₂ could be impressively lowered to 1 mol%. As shown in Table 4A, an array of *ortho*-substituted 4-iodo-2-pyridones reacted smoothly with methyl 2-bromobenzoate (**2A**) and **3a** to provide the arylated products (**4A−H**) in moderate to good yields (34−94%). Interestingly, the well-known *ortho*-effect of Catellani-type arylation[34] was not observed for *ortho*-unsubstituted 4-iodo-2-pyridone, since the bisarylated product **4I** was obtained in 69% yield. Then, the scope of arylation reagent was explored (Table 4B). Generally, aryl bromides with one *ortho*-electron-withdrawing group were good reagents, including ester (**2A−B**), amide (**2D−E**), nitro (**2K**) groups, even the reactive carboxylic acid (**2C**) and acetyl (**2F**) groups. Polysubstituted aryl bromides (**2G−K**) were also competent arylating reagents to deliver the desired 2-pyridones in 35−89% yields. As to the scope of terminating reagent, it was similar to the alkylation protocol (see Supplementary Fig. 5). Inspired by recent studies on Pd/chiral

norbornene asymmetric catalysis[65–67], we tested the synthesis of axially chiral 2-pyridone using enantiopure (+)-**N⁹** (99% *ee*)[66,67] as the mediator and sterically demanding 2,6-disubstituted aryl bromide **2L** as the arylating reagent. Pleasingly, the desired product **4A*** was obtained in 73% yield and excellent enantiomeric excess (97% *ee*), after minor modification of the reaction conditions (Table 4C and see Supplementary Table 8 for details).

*Two-component annulation, N-deprotection, and follow-up transformations of the obtained 2-pyridones.* Next, we focused on illustrating the synthetic utility of these protocols. First, an annulation process was explored based on a speculated two-component Catellani process (Fig. 3A). It was found that **1a** reacted with the bifunctional reagent **5** and **6** bearing a bromide and an olefin moiety, to afford the cyclized products **7** and **8** in excellent yields. Thus, it provided an efficient method for the assembly of bicyclic 2-pyridone derivatives[68–70]. Then, *N*-deprotection of the obtained various 2-pyridones were performed to set the stage for further manipulations (Fig. 3B, C). For example, *N*-Bn deprotection of **4l′** proceeded smoothly under the catalytic hydrogenation conditions to deliver **9** in 77% yield[71]. The *N*-PMB group of **4g** was readily removed in heated CF₃CO₂H to afford **10**[72], which could be quickly transformed into a OTf-substituted pyridine derivative (**11**) after treatment with Tf₂O and pyridine[73]. In addition, BBr₃ mediated *N*-MOM deprotection of **4d** took place at a low

**Table 4 Reaction scope of *ortho* C–H arylation.**

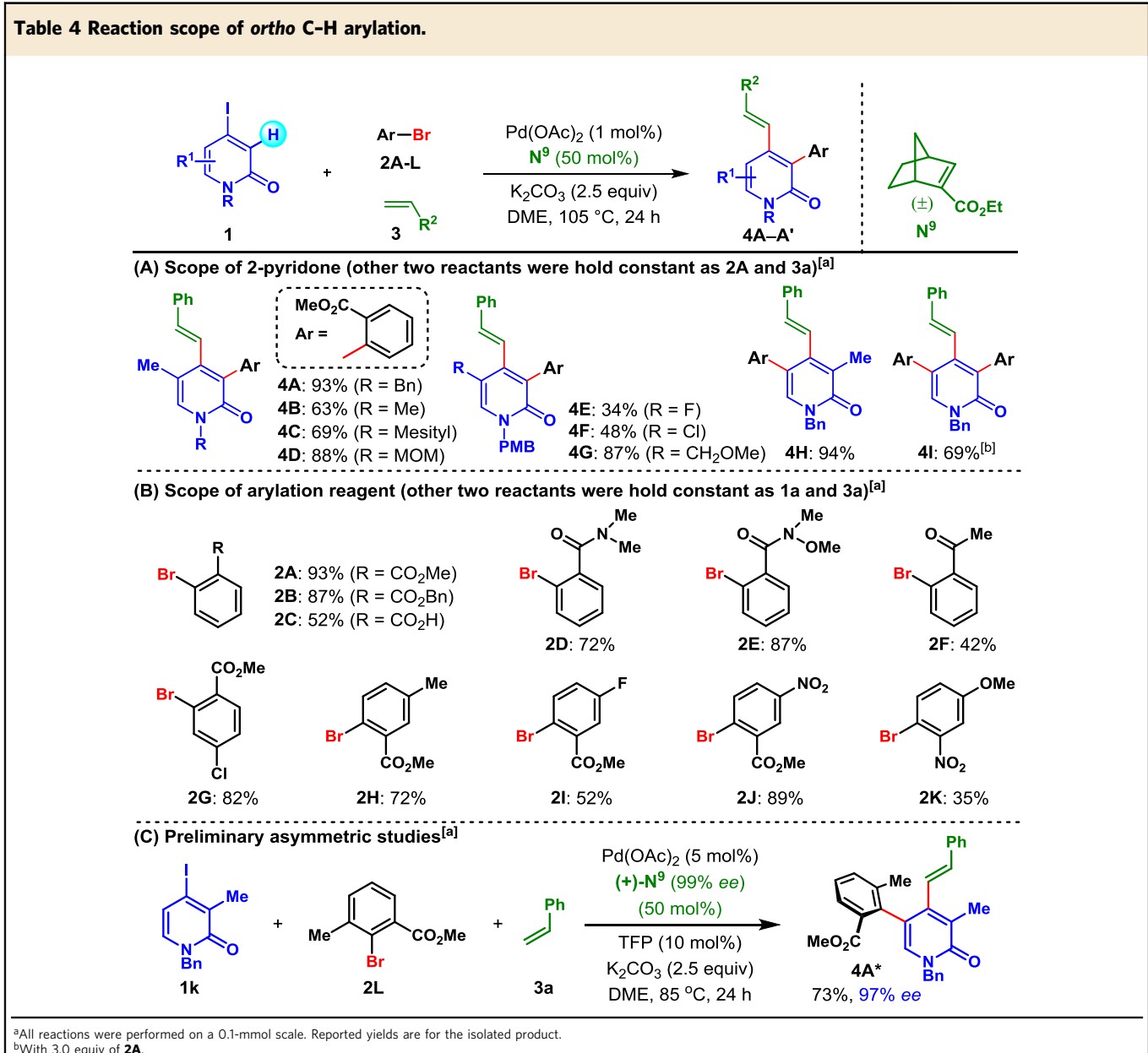

temperature[74], and the obtained intermediate **12** readily led to pyridine-fused lactone **13** through a facile two-step procedure[75]. Interestingly, BBr3 mediated *N*-MOM deprotection of the *ortho* C–H arylation product **14** provided two products: the normal one **15** in 27% yield and the methyl benzoate hydrolyzed product **16** in 42% yield, which were further transformed into complex pyridine derivatives **17** and **18** respectively in just one step with good yields.

*Diversity-oriented functionalization of uracils.* Additionally, the successful functionalization of iodinated uracil (**4m**, Table 2A) and biology-oriented synthesis (BIOS)[76] philosophy prompted us to further uncover the inherent value of our method by synthesizing biorelevant uracil and thymine derivatives (Fig. 4). Delightfully, with readily available iodinated uracils **1m** and **1n** as substrates, **2d** and **2A** as the electrophilic reagents, the 6-deuterated thymine derivative (80% deuteration) (**19**), and other five novel thymine analogs and derivatives (**20−24**) were facilely prepared in moderate yields, using *ipso*-hydrogenation (**19**),

-Suzuki (**20**), -Heck (**21−22**), or -Sonogashira termination (**23−24**) respectively. Furthermore, the reaction of **1m** with benzyl choride and methylboronic acid afforded **25** in 50% yield, resulting in a one-step formal synthesis of anti-HIV-1 agent **26** (previous method required 6 steps with only 6% overall yield)[77]. These results revealed the modularity of our method in diversity-oriented synthesis of useful uracil and thymine derivatives, which will be very attractive for developing new antiviral agents (Fig. 4B)[78–80].

In summary, we have developed a palladium/norbornene cooperative catalysis enabled diversity-oriented functionalization of heterocycles 2-pyridone and uracil. The success of this research depends on the use of two unique norbornene derivatives as the mediator. Readily available alkyl halides/tosylate and aryl bromides are utilized as *ortho*-alkylating and -arylating reagents, respectively. Widely accessible *ipso*-terminating reagents, including H/DCO2Na, boronic acid/ester, terminal alkene and alkyne are compatible with this protocol. Thus, a large number of useful derivatives of these heterocycles,

**Fig. 3 Two-component annulation, *N*-deprotection and follow-up transformations of the obtained 2-pyridones. A** Two-component annulation. **B** *N*-deprotection and follow-up transformations of the obtained 2-pyridones **4l'**, **4g**, and **4d**. **C** *N*-deprotection and follow-up transformations of the obtained 2-pyridone **14**.

including deuterium/CD$_3$-labeled 2-pyridones, bicyclic 2-pyridones, 2-pyridone-fenofibrate conjugate, axially chiral 2-pyridone (97% *ee*), as well as uracil and thymine derivatives, can be quickly prepared in a predictable manner, which will be very attractive for developing new generation of EZH2 inhibitors and antiviral agents. This work constitutes not only a nice extention of the Catellani reaction, but also a valuable addition to the toolbox of medicinal chemists

## Methods

**General procedure for *Ortho*-alkylating of 2-pyridone**. To an oven-dried Schlenk tube equipped with a magnetic stir bar were added Pd(OAc)$_2$ (5 mol%), norbornene derivatives **N$^8$** (0.05 mmol, 50 mol%), alkenyl iodide **1** (0.12 mmol, 1.2 equiv) and potassium carbonate (0.25 mmol, 2.5 equiv), and anhydrous 1,4-dioxane (1 mL) in the glove box. Then alkylating reagent **2** (0.15 mmol, 1.5 equiv) and terminating reagent **3** (0.1 mmol, 1.0 equiv) were added, and the mixture was heated to 105 °C. After completion of the reaction (monitored by TLC (thin layer chromatography), the mixture was cooled to r.t., filtered through a thin pad of celite, eluting with EtOAc

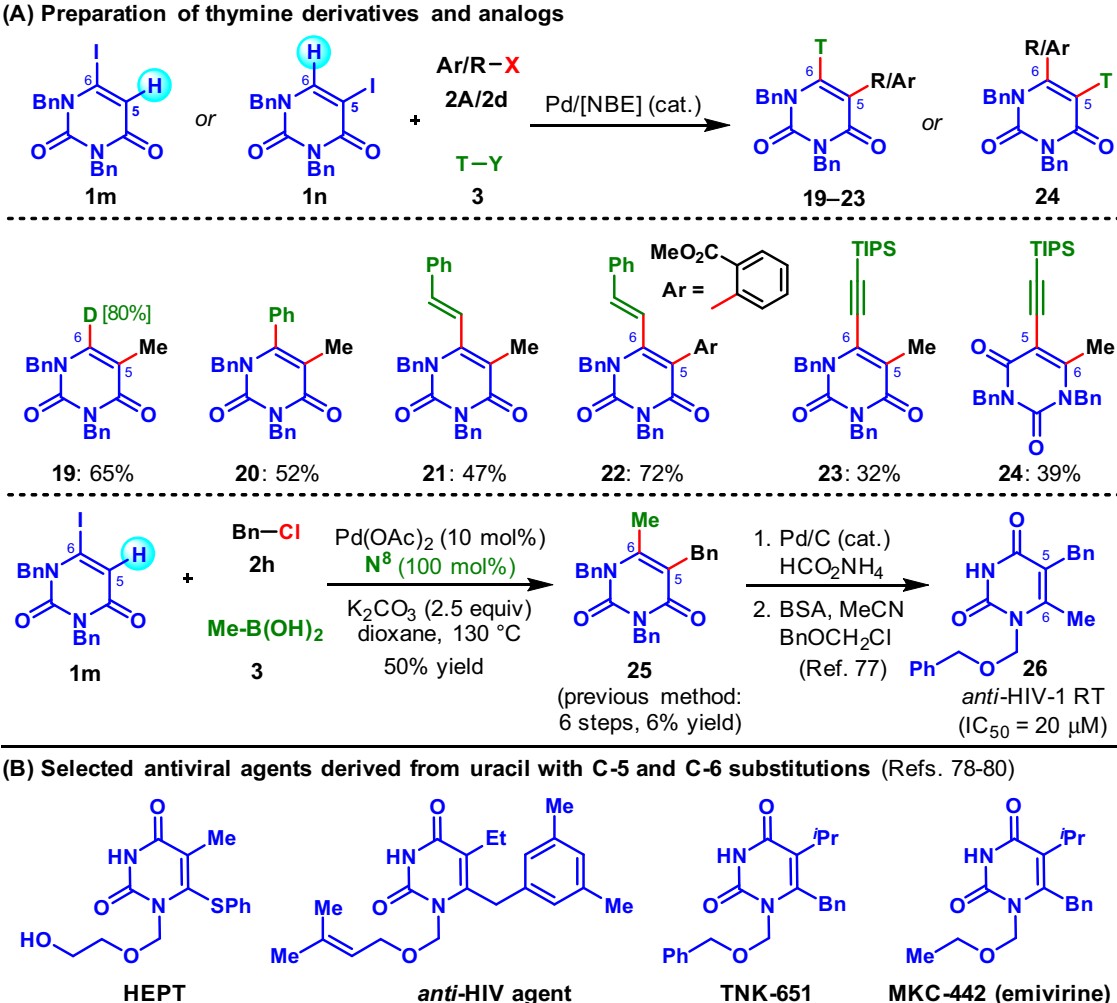

**Fig. 4 Diversity-oriented functionalization of uracils. A** Preparation of thymine derivatives and analogs. **B** Selected antiviral agents derived from uracil with C-5 and C-6 substitutions.

(10 mL), and the combined filtrate was concentrated in vacuo. The residue was directly purified by column chromatography on silica gel or purified by PTLC (preparative thin layer chromatography) to give the desired product.

**General procedure for *Ortho*-arylation of 2-pyridone**. To an oven-dried Schlenk tube equipped with a magnetic stir bar were added Pd(OAc)$_2$ (1 mol%), norbor-nene derivatives **N**[9] (0.05 mmol, 50 mol%), alkenyl iodide **1** (0.1 mmol, 1.0 equiv) and potassium carbonate (0.25 mmol, 2.5 equiv), and anhydrous DME (1 mL) in the glove box. Then alkylating reagent **2** (0.15 mmol, 1.5 equiv) and terminating reagent **3** (0.15 mmol, 1.5 equiv) were added, and the mixture was heated to 105 °C. After completion of the reaction (monitored by TLC (thin layer chromatography), the mixture was cooled to r.t., filtered through a thin pad of celite, eluting with EtOAc (10 mL), and the combined filtrate was concentrated in vacuo. The residue was directly purified by column chromatography on silica gel or purified by PTLC (preparative thin layer chromatography) to give the desired product.

## Data availability
The authors declare that all relevant data supporting the findings of this study are available within the paper and its supplementary information files.

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

## Acknowledgements
We are grateful to the National Key Research and Development Program of China (No. 2016YFB0101203) and start-up funding from Wuhan University for financial support. We thank Profs. Wen-Bo Liu, Suming Chen, and YongJia Tong (Wuhan University) for sharing their instruments, Dr. Ze-Shui Liu, Yu Hua, and Lan Zhou for affording some of the substrates, Jinxiang Ye for repeating some of the experiments, Tao Yang for assistance with the preparation of the manuscript.

## Author contributions
Q.Z. conceived the idea, guided the project, and wrote the manuscript. Y.S., C.W., Q.G., C.L., L.L, X.Z., and S.L. performed the experiments and analyzed the data. Y.S., C.W., and H.-G.C. participated in the preparation of the manuscript.

## Competing interests
Q.Z., Y.S., and C.W. have filed two provisional patient applications (202010807097.2, 202010807122.7). All other authors declare no competing interests.
