## [Peer Review File · Nature Communications]

REVIEWER COMMENTS

Reviewer #1 (Remarks to the Author):

Few times a year, I am reviewing a manuscript with no problems. 2021 is starting well with the current one on a modification of the Catellani reaction to functionalize pyridone in an efficient manner.

I have only one comment and one question:

- the title is misleading for me: I would replace synthesis by functionalization since the reaction is starting from a pyridone.
- I notice that phosphine were not necessary for most of the reactions (probably replaced by dioxane or DME). But tri(2-furyl)phosphine was added in the enantioselective version of the reaction. Do you have a hypothesis to explain this? I see in table S11 that MeCN is blocking the reaction. Could it be some cooperativity between the modified norbornene and the phosphine that explain the good ee?

Reviewer #2 (Remarks to the Author):

The authors report a diversity-oriented synthesis of a library of difunctionalized pyridone and thymine products from 4-iodopyridone and iodouracil starting materials, respectively, via Pd/NBE cooperative catalysis. Their study involved a comprehensive survey of various ortho electrophiles (alkylation and arylation) and ipso-terminating reagents (alkenylation, hydrogenation, alkylation, arylation, and alkynylation), along with showing two ring annulation examples and an excellent preliminary result for asymmetric arylation. The large quantity of difunctionalized products prepared, over 70, highlights the utility of Pd/NBE cooperative catalysis towards making a library of complex molecules from relatively simple precursors.

However, the novelty of the transformation is diminished since the majority of the 4-iodopyridone substrates employed in this manuscript are quite similar to the 4-iodoquinolone substrates used by Yamamoto and co-workers in their 2018 Chemical Science report (ref. 43), which are mistakenly labeled as "4-iodo-quinones" in the introduction. It should also be noted that the partial aromaticity of the pyridone substrates, and the fact that they're typically used in excess, unfortunately diminishes the overall innovation of the transformation. Additionally, while the transformation of 1l to 15 for the formal synthesis of the anti-HIV-1 agent 16 is a great highlight of this method's power to rapidly increase molecular complexity towards the synthesis of useful intermediates from simple precursors, the inclusion of a more compelling synthetic application involving pyridones would increase the value of this work, as they are by far the primary focus of the article.

While I believe this is good, publishable work and will be of value to medicinal chemists, it is my recommendation that this manuscript be submitted to a more chemistry focused journal, as it is below the standard for Nature Communications in terms of novelty and innovation.

Reviewer #3 (Remarks to the Author):

Review: Diversity-Oriented Synthesis of 2-Pyridones and Uracils

The authors demonstrate that the Catellani reaction can be applied to iodinated 2-pyridones and uracils. While the finding is not overly surprising the scope is demonstrated to a high degree and many beautiful molecules are exemplified. A variety of bio-relevant heterocycles were capable of bis functionalization providing highly decorated molecules rapidly. Additionally, the asymmetric reaction they achieved was really nice. Overall, the work appears to be conducted with care and I would recommend this manuscript for publication though there are a couple points that need addressing.

1) The authors claim throughout the manuscript that pyridones are "non-aromatic". This is a factually incorrect statement. I could find references that indicate these systems are aromatic though they are less aromatic than benzene (they have approximately 35% of the aromaticity of

benzene). Additionally, the systems presented in the manuscript behave as aromatic systems. If the authors contend that they are non-aromatic this would require further discussion with appropriate references to make this claim. This fact, however, is not or should not be central to the main points of the manuscript. It seems to the reader that the crux of this claim is to bolster the importance of the transformation though this is not necessary as the work is good without the additional "challenge" of performing Catellani reactions on non-aromatic systems, which is not what is being presented here.

a. J. Chem. Soc., 1961, 859-866 (<https://doi.org/10.1039/JR9610000859>)

2) A very large percentage of the molecules presented are benzyl protected. In the context of complex or bioactive molecules this protecting group is not always trivial to remove. Additionally, many of the molecules that are prepared contain functional groups that are not compatible with the common strategies for benzyl group removal. It would be of benefit to the community and significantly strengthen this work if the authors could demonstrate removal of the benzyl group as well as demonstrate the chemistry with alternative protecting groups that are easier to remove. PMB was shown as an example however its removal was not demonstrated. Additionally Figure 3 shows a scheme where the benzyl group was removed though it was done so in a much simpler context when compared to the other molecules prepared in the manuscript and thus is not completely relevant.

3) The authors show in Figure 3, transformation 15 to 16, the preparation of an anti-HIV-1 molecule. It is unclear by the scheme if this reaction was performed by the authors or if it is more like a formal synthesis and the last step is a reference. This should be clarified in the text. If performed by the authors a yield should be provided in the scheme.

4) The authors claim in the conclusion that "novel norbornene catalysts" were used however the norbornene catalysts used are known and the authors reference them in the text. They should remove this statement as I think these factors preclude these norbornene catalysts from being referred to as novel.

5) Authors should take care to define abbreviations the first time they are used. For example, PTLC = preparative thin layer chromatography.

6) In the introduction the authors state that: "Owing to diversified and orthogonal dual-functionalization (ortho/ipso) of aryl halides, the Catellani reaction has become a versatile tool for quickly building aromatics library". I think stating that this reaction is general to aryl halides is misleading and more the exception than the rule. This chemistry almost only works with aryl iodides. This should be stated more clearly.

7) Authors should include mol % of catalyst used in the experimental.

8) The authors should check their references for correct numbering. For instance, reference 70 in the following sentence should be 71

- "Additionally, the successful functionalization of iodinated uracil (4l, Table 2A) and biology-oriented synthesis (BIOS)70 "

We thank the referees for their constructive criticisms that have helped us to improve the quality of the manuscript. The point-by-point responses to the referees's concerns are listed below, and the detailed changes can be found in the marked versions of the revised Manuscript and Supporting Information.

Reply to Referee 1

Critique: “The title is misleading for me: I would replace synthesis by functionalization since the reaction is starting from a pyridone.”

Response: Thanks for this valuable suggestion. Now, the title was changed as “Diversity-Oriented Functionalization of 2-Pyridones and Uracils”.

Critique: “I notice that phosphine were not necessary for most of the reactions (probably replaced by dioxane or DME). But tri(2-furyl)phosphine was added in the enantioselective version of the reaction. Do you have a hypothesis to explain this? I see in table S11 that MeCN is blocking the reaction. Could it be some cooperativity between the modified norbornene and the phosphine that explain the good ee?;

Response: We thank the referee for this insightful comment. We think the addition of tri(2-furyl)phosphine (TFP) may increase the catalytic reactivity and efficiency since the sterically demanding 2,6-disubstituted aryl bromide **2L** is less reactive as compared to the ones (**2A-K**) in Table 4B. In addition, the reaction temperature of the enantioselective variant is just 85 °C, 20 °C below the standard temperature in Table 4A-B, which is an obvious support of our hypothesis. However, we are currently not quite sure about its exact effect on the enantioselectivity of this reaction. Additional experiments are needed to elucidate this issue, which will be our next focus.

Reply to Referee 2

Critique: “However, the novelty of the transformation is diminished since the majority of the 4-iodopyridone substrates employed in this manuscript are quite similar to the 4-iodoquinolone substrates used by Yamamoto and co-workers in their 2018 Chemical Science report (ref. 43), which are mistakenly labeled as “4-iodo-quinones” in the introduction. It should also be noted that the partial aromaticity of the pyridone substrates, and the fact that they're typically used in excess, unfortunately diminishes the overall innovation of the transformation.”

Response: Thanks for this criticism. We are sorry for the typos. Accordingly, the “4-iodo-quinones” has been corrected as “4-iodoquinolones” and “non-aromatic” has been corrected as “partially aromatic”. In the *ortho* arylation reactions, only 1 equivalent of 4-iodo-2-pyridone substrates were employed. In the *ortho* alkylation reactions, 1.2 equivalent of 4-iodo-2-pyridone substrates were used to achieve excellent yields. Actually, when 1 equivalent of 4-iodo-2-pyridone substrates were used, the desired products can also be obtained in good yields (eg. **4a** 83% NMR yield, Table 1, entry 10).

As to the comparison of Yamamoto's work with ours, we provided the details as follows. The Yamamoto group reported a **two-component** Catellani-type annulation reaction, involving heteroaromatic 4-iodo-quinolones (5 cases) and 4-iodo-coumarin

(1 case) as the substrate. It is worth mentioning that 7-8 steps are required for the preparation of 4-iodo-quinolone substrates and only tricyclic products are afforded, which may prevent the wide application of this chemistry (Ref 43, Figure 1A).

Our work disclosed an efficient **three-component** Catellani reaction for the preparation of structurally diversified 2-pyridones and uracils, utilizing unsymmetrical norbornene derivative (N^8/N^9) as the mediator. Readily accessible iodinated 2-pyridones and uracils were used as the substrates. Remarkably, diversity-oriented functionalization of 2-pyridones and uracils were realized (79 examples reported), including deuterium/fluoro-labeled 2-pyridones, bicyclic 2-pyridones, 2-pyridone-phenofibrate conjugate, as well as uracil and thymine derivatives (the potential of this chemistry for constructing 2-pyridone and uracil libraries is enormous, considering the broad scope of alkylating/aryllating reagents and terminating reagents). More importantly, this chemistry will be very attractive for developing new antiviral agents, owing to its ability for the facile synthesis of diversified uracil and thymine derivatives. Thus, this study is of significant value to pharmaceutical industry. (Figure 1B).

As we know, the norbornene mediator plays a pivotal role in Catellani-type reactions. In our study, a significant mediator effect was observed, which led us to identify the optimal mediators N^8 and N^9 for this challenging three-component Catellani reaction. This represents an important step forward as compared to previous work in this area. Furthermore, by utilizing the chiral N^{9*} (50 mol%, 99% *e.e.*), a stereoselective synthesis of axially chiral 2-pyridone (97% *e.e.*) was achieved. Different from our work, Yamamoto's studies (Ref 43) were performed just using 1.0-2.0 equivalents of simple 2-norbornene (NBE) as the mediator, which restricted their studies to two-component annulation reactions.

A) Yamamoto's work (Ref 43: Yamamoto, Y. *et al. Chem. Sci.* 2018, 9, 1191)

B) Our work (79 examples reported)

From Y. Yamamoto, T. Murayama, J. Jiang, T. Yasui and M. Shibuya, *Chem. Sci.*, 2018, 9, 1191 licensed under a CC BY 3.0 (<https://creativecommons.org/licenses/by/3.0/>) license.

Figure 1: The comparison of Yamamoto's work with ours

At this stage, I think it is evident that our work is very different from their studies, in terms of different substrates scope, the used NBE mediators, the corresponding transformations and products as well as synthetic applications.

Critique: “Additionally, while the transformation of **11** to **15** for the formal synthesis of the anti-HIV-1 agent **16** is a great highlight of this method’s power to rapidly increase molecular complexity towards the synthesis of useful intermediates from simple precursors, the inclusion of a more compelling synthetic application involving pyridones would increase the value of this work, as they are by far the primary focus of the article.”

Response: We thank the referee for this constructive suggestion. Accordingly, we performed some compelling synthetic applications involving the synthesis of complex polysubstituted pyridines from 2-pyridones. The details are shown in Scheme 1 in the revised manuscript.

Reply to Referee 3

Critique: “The authors claim throughout the manuscript that pyridones are “non-aromatic”. This is a factually incorrect statement. I could find references that indicate these systems are aromatic though they are less aromatic than benzene (they have approximately 35% of the aromaticity of benzene). Additionally, the systems presented in the manuscript behave as aromatic systems. If the authors contend that they are non-aromatic this would require further discussion with appropriate references to make this claim. This fact, however, is not or should not be central to the main points of the manuscript. It seems to the reader that the crux of this claim is to bolster the importance of the transformation though this is not necessary as the work is good without the additional “challenge” of performing Catellani reactions on non-aromatic systems, which is not what is being presented here.”

Response: Thanks for this helpful suggestion. We apologize for the mistake about the concept of aromaticity. We totally agree with the referee’s point. Accordingly, the “non-aromatic” has been corrected as “partially aromatic” in the revised manuscript.

Critique: “A very large percentage of the molecules presented are benzyl protected. In the context of complex or bioactive molecules this protecting group is not always trivial to remove. Additionally, many of the molecules that are prepared contain functional groups that are not compatible with the common strategies for benzyl group removal. It would be of benefit to the community and significantly strengthen this work if the authors could demonstrate removal of the benzyl group as well as demonstrate the chemistry with alternative protecting groups that are easier to remove. PMB was shown as an example however its removal was not demonstrated. Additionally Figure 3 shows a scheme where the benzyl group was removed though it was done so in a much simpler context when compared to the other molecules prepared in the manuscript and thus is not completely relevant.”

Response: We thank the referee for this constructive suggestions. We identified PMB

and MOM to be good alternative *N*-protecting groups of 2-pyridones. Following the reported deprotection conditions, the removal of the three type of protecting groups all proceeded smoothly to give the desired products in good yields, paving the way for further manipulations. The details are shown in Scheme 1 of the revised manuscript.

Critique: “The authors show in Figure 3, transformation **15** to **16**, the preparation of an anti-HIV-1 molecule. It is unclear by the scheme if this reaction was performed by the authors or if it is more like a formal synthesis and the last step is a reference. This should be clarified in the text. If performed by the authors a yield should be provided in the scheme.”

Response: We thank the referee for the suggestion. The preparation of **26**, an anti-HIV-1 molecule, is a formal synthesis, we have clarified it in the revised manuscript.

Critique: “The authors claim in the conclusion that “novel norbornene catalysts” were used however the norbornene catalysts used are known and the authors reference them in the text. They should remove this statement as I think these factors preclude these norbornene catalysts from being referred to as novel.”

Response: We thank the referee for this suggestion. The statement has been changed accordingly in the revised manuscript.

Critique: “Authors should take care to define abbreviations the first time they are used. For example, PTLC = preparative thin layer chromatography.”

Response: The abbreviations have been defined accordingly.

Critique: “In the introduction the authors state that: “Owing to diversified and orthogonal dual-functionalization (ortho/ipso) of aryl halides, the Catellani reaction has become a versatile tool for quickly building aromatics library”. I think stating that this reaction is general to aryl halides is misleading and more the exception than the rule. This chemistry almost only works with aryl iodides. This should be stated more clearly.”

Response: We thank the referee for the comment. Accordingly, we changed the statement to “...dual-functionalization (ortho/ipso) of aryl halides (mainly aryl iodides)...”. Please see the details in the revised manuscript.

Critique: “Authors should include mol % of catalyst used in the experimental. The authors should check their references for correct numbering. For instance, reference 70 in the following sentence should be 71. Additionally, the successful functionalization of iodinated uracil (4l, Table 2A) and biology-oriented synthesis (BIOS)70.”

Response: Corrected as suggested.

REVIEWERS' COMMENTS

Reviewer #1 (Remarks to the Author):

I checked the revisions made by the authors and I think they improve the manuscript, which can be now published.

Reviewer #2 (Remarks to the Author):

In the revised submission, technical aspects of the work have indeed been improved. However, the novelty of the work remains unaltered. This reviewer feels surprised that other two reviewers seem to have omitted two important prior works, particularly Yamamoto's 2018 Chemical Science publication, where similar substrates have been demonstrated.

Lautens, M. Preparation of Annulated Nitrogen-Containing Heterocycles via a One-Pot Palladium-Catalyzed Alkylation/Direct Arylation Sequence. *Org. Lett.* 8, 2043-2045 (2006).

Yamamoto, Y. The Vinylogous Catellani Reaction: A Combined Computational and Experimental Study. *Chem. Sci.* 9, 1191-1199 (2018).

In addition, these prior arts lack proper citation and description in the introducing text and figure. While the authors try to spin out a different application or concept, the chemical transformation is not fundamentally different.

Since the transformation is not new, we may next ask: is there any new mechanistic insight obtained in this work? The answer is clearly No. This work shares the identical reaction pathway as the prior works.

Furthermore, we may ask: is there any new catalyst or new reaction conditions identified in this work? Unfortunately, the answer is still No. The norbornene catalysts used in this work has been previously developed by others and used in similar contexts.

Overall, based on the above analysis, while this is a nice extension of the Catellani reaction and the paper is well written and packaged, both the novelty and significance of this work are much below the standard of Nature Communications.

Reviewer #3 (Remarks to the Author):

The authors did well to satisfactorily respond to the reviewer's comments. I recommend publication.